👁 | Open Peer Review | Microbial Pathogenesis | Research Article

# Unraveling the mouse model of *Staphylococcus aureus* bacteremia and sepsis: a systematic approach to better characterize host/pathogen interactions

Serena Vastola,[1,2,3] Marco Tortoli,[1] Stefania Torricelli,[1] Michela Brazzoli,[1] Marco Maria D'Andrea,[2] Emiliano Chiarot[1]

**ABSTRACT** *Staphylococcus aureus* is a pathobiont whose primary human reservoirs are nares, pharynx, intestines, and skin. When specific conditions in the host are altered, it can cause a wide variety of human diseases, including bacteremia and sepsis. Preclinical *in vivo* models mimicking the most severe *S. aureus* infections in humans have been used to develop treatments against this pathogen. This study aims to better characterize a murine model of *S. aureus* bacteremia and sepsis, offering a new and more comprehensive view of the complex interactions between *S. aureus* and the host while better reflecting human disease dynamics. We investigated the kinetics of bacteria in blood, kidneys, and liver after infection with four strains representative of epidemiologically relevant *S. aureus* clonal lineages. After intravenous infection, bacteria progress through three major pathogenesis phases: (i) colony-forming units counts in blood decrease rapidly within 1–2 h as bacteria are captured by the liver, the first line of defense against blood-borne bacteria; (ii) mice begin to show signs of acute disease, and bacteria disseminate to the kidneys where they grow quickly, reaching the peak in 1–2 days; (iii) bacteria establish an equilibrium with the host, forming abscesses in the kidneys while persisting in low numbers in the blood. These phases are common to all the tested *S. aureus* strains, although some strain-specific peculiarities have also been identified. Our findings could help improve understanding of host-pathogen interactions in *S. aureus* infections and their implications for human health, potentially laying the groundwork for developing novel preventive and therapeutic strategies.

**IMPORTANCE** Our work provides new insights into the interaction between *Staphylococcus aureus* and the host in a mouse model of bloodstream infection and sepsis. We found similarities between findings in the mouse model and human disease, underscoring the importance of using this laboratory host to study new therapeutic and preventive interventions. The comprehensive approach we used, utilizing several epidemiologically relevant *S. aureus* clones and two distinct mouse strains, enhances the relevance of our results and sheds light on the complex interaction between this human pathogen and a widely used laboratory research host. We believe this approach could also be useful for studying *S. aureus* infections in different animal models of disease.

**KEYWORDS** murine model, bloodstream infection, MRSA, pathogenesis

**Peer Reviewers** Basel H. Abuaita, Louisiana State University School of Veterinary Medicine, Baton Rouge, Louisiana, USA; Sreekanth Reddy Basireddy, Kurnool Medical College, Andhra Pradesh, India

Address correspondence to Emiliano Chiarot, emiliano.x.chiarot@gsk.com.

S.V. is a Ph.D. student of Università Roma Tor Vergata, Italy, and is supervised by GSK. M.T., S.T., M.B., and E.C. are employees of the GSK group of companies. M.B. and E.C. report ownership of GSK shares. The remaining author has no conflict of interest to declare.

*S*taphylococcus aureus was first described in 1880 by Alexander Ogston, who isolated this bacterium from an infected wound and exposed the role of the pathogen in the etiology of the pyogenic abscesses (1). *S. aureus* is a widespread pathobiont with primary human reservoirs in the nares, pharynx, intestines, and skin (2–5). It is common for healthy adults to carry *S. aureus*, with about one-third of the global population being persistently colonized; this percentage increases to roughly 90% when intermittent

carriers are included (6, 7). Since its first discovery, *S. aureus* has settled as a threatening pathogen and a critical public health concern, listed as a priority pathogen by the World Health Organization (8, 9). It can cause skin and soft tissue infections (SSTIs) but also necrotizing pneumonia and invasive syndromes, such as necrotizing fasciitis, osteomyelitis, septic thrombophlebitis, bacteremia, and severe sepsis (10). Additionally, resistance to multiple antibiotics, especially to methicillin and conventional β-lactams, is common and makes treatments particularly difficult (11). Indeed, infections caused by methicillin-resistant *S. aureus* (MRSA) result in increased morbidity, mortality, length of hospital stay, and costs compared to those caused by methicillin-susceptible *S. aureus* (12). Therefore, new antibiotics or alternative preventive/therapeutic strategies are urgently needed to tackle *S. aureus* showing multidrug resistance and high virulence (13).

The availability of robust preclinical models that mimic the most severe *S. aureus* infections in humans is a first important step to evaluate the effectiveness of potential alternative therapeutic strategies. Mouse models of bacteremia/sepsis following intravenous administration of bacteria have been developed more than 30 years ago, and they are still largely used in research to evaluate and develop new preventing and therapeutic treatments (14, 15). Nevertheless, they have not been deeply characterized, and differences between disease progression between humans and mice are not well understood. For instance, when *S. aureus* reaches the blood in humans, there is a phase of rapid growth (bacteremia), and bacteria can also easily spread to all the organs (sepsis) (16, 17). On the other hand, after intravenous administration of *S. aureus* in mice, infection is promptly controlled in the blood, even if this does not affect the ability of bacteria to reach organs and tissues where they organize in abscesses and could survive for a long period (18–20).

In the present work, we tried to better understand and characterize a murine model of bloodstream infection with the aim to offer a new and more comprehensive view of the complex interactions between *S. aureus* and the host. In particular, mice were infected with four different strains representative of epidemiologically relevant clones, and their impact on disease progression in animals has been explored. We investigated the kinetics of bacteria in blood to follow the "bacteremia phase" and in different organs to follow the "sepsis phase." Moreover, we compared the infection in two different mouse strains to evaluate the impact the host genetic background can have in the host-pathogen interaction. Overall, these findings can illuminate mechanisms of disease development and host response, potentially leading to novel approaches against systemic *S. aureus* infections.

## MATERIALS AND METHODS

### Bacterial strains and preparation of inoculum for infection

Four strains of *S. aureus*, representatives of epidemiologically relevant clonal lineages, were chosen for this *in vivo* model: LAC (USA300 – CC8), MW2 (USA400 – CC1), μ50 (USA100 – CC5), and TW20 (ST239 – CC8) (Table 1). LAC (USA300 – CC8), MW2 (USA400 – CC1), and μ50 (USA100 – CC5) were kindly provided by the University of Chicago (Chicago, Illinois, United States); TW20 (ST239 – CC8) was kindly provided by Guy's and St Thomas' NHS Foundation Trust (London, United Kingdom).

Bacteria were grown in tryptic soy broth (TSB) at 37°C with agitation (250 rpm) until the early exponential phase (Optical Density - $OD_{600}$= 2) then diluted 1:1 in a freezing solution, composed of phosphate-buffered saline (PBS, Invitrogen pH 7.4) + 10% bovine serum albumin (BSA, Sigma) + 10% L-glutamic acid monosodium salt hydrate (MSG, Sigma). Aliquots were stored at −80°C in cryovials until use. For the inoculum preparation, bacteria were grown to the early exponential phase, corresponding to approximately $10^9$ colony-forming units (CFU)/mL, as previously reported (31). Bacteria were then washed once in sterile PBS and diluted to obtain the desired concentration for infection.

**TABLE 1** Features of the selected *S. aureus* strains

| *S. aureus* strains | Sequence type (ST) | Clonal complex (CC) | Origin | Isolation | General features | References |
|---|---|---|---|---|---|---|
| LAC (USA300 -CC8) | ST8 | CC8 | Skin infection | 2002 | *S. aureus* USA300 clones primarily cause skin infections and account for up to 98% of all methicillin-resistant *S. aureus* (MRSA) skin and soft tissue infections (SSTIs) in the USA. In addition, USA300 can also cause more invasive diseases such as bacteremia, endocarditis, and necrotizing fasciitis. | (21–23) |
| MW2 (USA400 – CC1) | ST1 | CC1 | Septicemia and septic arthritis | 1998 | MW2 (USA400 clone) is a typical community-acquired strain of MRSA that was isolated in 1998 in North Dakota, USA. The MW2 strain is susceptible to many antibiotic classes, apart from β-lactam antibiotics. | (24, 25) |
| µ50 (USA100 – CC5) | ST5 | CC5 | Surgical wound infection | 1997 | *S. aureus* USA100 clones are the predominant lineage colonizing human nares in the USA. Notably, USA100 strains are a leading cause of invasive disease among MRSA isolates and represent the majority of vancomycin-resistant/intermediate *S. aureus* isolates. There is no specific association known between the USA100 clone and particular infections. | (26–28) |
| TW20 (ST239 – CC8) | ST239 | CC8 | Vascular Access Device - Related Bacteraemia | 2004 | Acquisition of TW20 (ST239) was found to be four times more likely associated with bacteremia than the acquisition of other epidemiologically relevant MRSA strains. It is more frequently isolated from vascular access device cultures but less frequently from carriage sites. It exhibits a broad range of antibiotic resistances (e.g., penicillins, methicillin, erythromycin, ciprofloxacin, gentamicin, neomycin, trimethoprim, and tetracycline). | (29, 30) |

## Animal care and ethical statements

Animal husbandry and experimental procedures were ethically reviewed and carried out in accordance with European Directive 2010/63/EU, Italian Decree 26/2014 and GSK Vaccines' Policy on the Care, Welfare and Treatment of Animals, and were approved by the Italian Ministry of Health (authorization n° 520/2023-PR). Animals were kept in an AAALAC-accredited facility, as previously described (32). Briefly, animals were randomly distributed in different experimental groups in individually ventilated cages (IVC). Each animal was identified by an individual tattoo on the tail at the end of an acclimation period of at least 5 days. The animal room conditions were the following: temperature 21°C (± 3°C), relative humidity 50% (range 30%–70%) and 12 h/12 h light/dark cycle. Pressure, temperature, and relative humidity were recorded continuously by room probes, while the IVC system recorded the individual motors' performance. The light cycle setting was ensured by a validated alarm system.

## Mouse model of bloodstream infection

The mouse model of bloodstream infection was performed as depicted in Fig. S1. Eight-week-old female specific pathogen-free (SPF) CD-1 or C57BL/6N were purchased from Charles River Italy and injected with $0.5–1.0 \times 10^7$ CFU/100 µL of *S. aureus* into the tail vein. Mice were observed daily for the development of clinical signs of disease and body weight loss for up to 9 days (CD-1 mice) or 2 days (C57BL/6N mice) after infection. Mice were euthanized if they reached pre-established scoring linked to humane endpoints, in accordance with the approved ministerial authorization. Euthanasia was carried out through cervical dislocation, followed by confirmation of death through assessment of clinical signs, as per national legislation and internal policies. The endpoints were based on a series of parameters that included percentage of body weight change and other observations and measurements indicating pain or distress, as previously reported (33–35). Specifically for this model, we recognized four different stages of body weight loss correlating with the development of disease: no disease (>95% of the initial body weight measured before infection), mild disease (90–95% of the initial weight), moderate disease (80%–90% of the initial weight), and severe disease (<80% of the initial weight). Besides body weight, mice were also observed to monitor the level of ruffled fur, presence of kyphosis, and activity levels, using the following

scoring system: (i) scruffy coat: mild = 1, moderate/severe = 2; (ii) kyphosis: presence = 1; (iii) normal activity level = 0; slightly decreased activity level = 1; moderately suppressed activity level = 2; suppressed activity levels (stationary with occasional investigative movements) = 3; no activity (mouse is stationary) = 4. The combination of body weight loss >20% and decreased activity level scores 3 or 4 was considered a final humane endpoint. Other signs of disease that presented less frequently and consistently as mild to moderate were considered less appropriate to address and intercept welfare aspects relevant and useful to the model and were therefore not used as humane endpoints. At specified time points after infection, blood was withdrawn from the mandibular vein and collected in heparin (Eparina Vister 5000 UI/mL) at a final concentration of 500 UI/mL or above. For blood sampling at intermediate time points, the acceptable quantity of blood sampling was calculated as 10% of circulating blood volume. We determined total blood volume from the average value of body weight specific to mouse strain and age, considering that it is accounted for approximately 58.5 mL/kg (Blood sampling: Mouse | NC3Rs). Mice were euthanized at predetermined time points, as previously described, and kidneys and liver were excised, collected in PBS, and homogenized using the gentleMACS Octo Dissociator (Miltenyi Biotec) following manufacturer's procedures. The resulting suspension was filtered using 70-µm cell strainers (Falcon), diluted (7× 10-fold dilutions on 96-well plates), and 2× 10-µL drops have been plated onto selective and differential chromogenic medium (BD, ChromAgar MRSA II) to evaluate *S. aureus* bacterial load. To eliminate potential bias, samples were randomized to each treatment using an internally developed randomization software. Blinding was applied for enumerating CFU obtained from homogenized organ specimens. Samples were unblinded at the end of colonies enumeration and associated to the identification number. CFU were expressed as CFU/mL for blood or CFU/organ for kidneys or liver.

## Statistical analysis

The Kruskal–Wallis test and uncorrected Dunn's post-test were used to assess differences among distinct groups at the same time point. A *P*-value ≤ 0.05 was considered significant for all analyses. Statistical analyses were performed using GraphPad Prism 9 software.

## RESULTS

### *S. aureus* strains belonging to distinct clonal lineages showed differences in both the magnitude and the kinetics of disease progression in mice infections

We evaluated the ability of the selected strains to infect animals after intravenous administration with outbred mice chosen for the first set of experiments to better mimic the genetic diversity of the human population. Eight-week-old CD-1 female mice were intravenously infected with $1 \times 10^7$ CFU of one of the selected strains and then followed for up to 9 days post-infection (p.i.). Based on published data, this infective dose was chosen to (i) allow all the infected mice to reach the 9-day observation period without needing to be sacrificed upon reaching the predefined humane endpoints; (ii) explore possible differences in infectivity among the selected strains (36, 37). A limited number of *S. aureus* clonal lineages, including CC5 and CC8, are commonly prevalent in MRSA bloodstream infections worldwide (38). In some countries, such as Japan, the epidemiological scenario also includes MRSA of CC1 acting as either colonizers or invasive strains (39). On these grounds, we selected four strains, LAC (USA300 – CC8), MW2 (USA400 – CC1), µ50 (USA100 – CC5), and TW20 (ST239 – CC8), to evaluate potential clone-dependent differences in a mouse model of bacteremia and sepsis. Body weight loss is a valuable predictor of disease progression for this model of infection and, accordingly, it was monitored daily to assess the general health status of the animals (40). Interestingly, mice infected with different strains showed different levels of body weight loss and, therefore, of disease severity. In general, we concluded that almost all the mice infected

with 3 out of 4 strains showed a decrease in body weight during the first 4–7 days after infection and then reached a steady state, but the magnitude and kinetics of this body weight loss trend appeared different among the strains (Fig. 1). Notably, *S. aureus* μ50 (USA100 – CC5) was the least prone to provoke disease among the selected strains since almost all the mice did not lose body weight during the 9-day observation period. Conversely, among the other three strains, the most virulent was MW2 (USA400 – CC1). Indeed, all the animals infected with this strain presented moderate to severe signs of disease at the end of the observation, with a median body weight loss higher than 20%. These groups of mice also showed other clinical signs, including decreased activity levels, kyphosis, and ruffled fur, although none of these infected mice needed to be sacrificed based on our predefined humane endpoints (data not shown). Mice infected with CC8 strains exhibited an intermediate condition: upon reaching the "steady state condition," animals showed mild to moderate signs of disease, with 2 out of 8 animals infected with the LAC (USA300 – CC8) strain being severely sick. To conclude, different strains can exhibit varying infectivity affecting the magnitude and kinetics of disease progression.

## *S. aureus* can persist in mouse blood for at least 9 days following infection

Previous data showed that intravenous infection with *S. aureus* can result in a very short initial phase in which bacteria remain in the blood (bacteremia phase) and a second phase (sepsis phase) when bacteria reach organs and tissues and form abscesses (18, 19). To better characterize these two phases, we enumerated bacteria in the blood of infected animals for up to 9 days after infection by plating 50 μL (early time points, up to 4 hours p.i.) or 200 μL (other time points) of freshly withdrawn heparinized material on MRSA selective plates. This allowed us to reduce the limit of detection to 5–20 CFU/mL of blood. In addition to these experiments performed to follow the bacteremia phase, we also tracked bacteria migrating to different organs to monitor the kinetics of the infection. Overall, we observed that shortly (1 h) after infection, only 0.03%–0.1% of the initial bacteria inoculum were still present in the blood, despite the fact that all the animals received $1 \times 10^7$ CFU intravenously (Fig. 2). Interestingly, during these early phases of the disease, CFU counts in the blood of animals infected with *S. aureus* TW20 (ST239 – CC8) were threefold higher compared with the other strains ($P < 0.01$), suggesting that this strain could survive better in the blood (Fig. 2; Table 2). Between 4 and 24 h after intravenous administration of bacteria, the concentration of *S. aureus* in mouse blood progressively decreased. This was particularly evident for the μ50 (USA100 – CC5) strain, which was completely cleared in more than 50% of infected mice within 2 days and in almost all mice within 9 days (Fig. 2). For the other strains, this decrement was more gradual; 9 days after infection, 43%–75% of infected mice still had bacteria in the blood, with the highest percentages for LAC (USA300 – CC8) and MW2 (USA400 – CC1) strains.

## *S. aureus* rapidly disseminates into organs within the first hours of infection

A few hours after infection, bacteria had already spread to the organs. Indeed, *S. aureus* was found in kidneys at the earliest time point investigated (2 h) and then replicated quickly over time. Bacteria reached a plateau in 48 h (Fig. 2), ranging from approximately $1–7 \times 10^3$ CFU/organ at 2 h p.i. to approximately $10^5–10^7$ CFU/organ at 48 h p.i. This trend was observed for three out of four tested strains, while in animals infected with *S. aureus* μ50 (USA100 – CC5), bacteria accumulated poorly in the kidney throughout the observation period. Interestingly, the dynamic of kidney localization was accelerated for the MW2 (USA400 – CC1) strain (Fig. 2; Table 2). The liver is a key player governing the kinetics and dynamics of *S. aureus* infection (41). When bacterial counts in the liver were analyzed over time, there were very few differences among the different strains (Fig. 2) at 2 h post-infection. A few hours after infection, all the animals presented high and comparable levels of bacteria in the liver (approximately $0.5–2.0 \times 10^6$ CFU/mouse, independent of the bacterial strain used). However, this count rapidly decreased by approximately 2 logs by 24 h p.i., reaching a plateau ($10^4$ bacteria/organ), which was

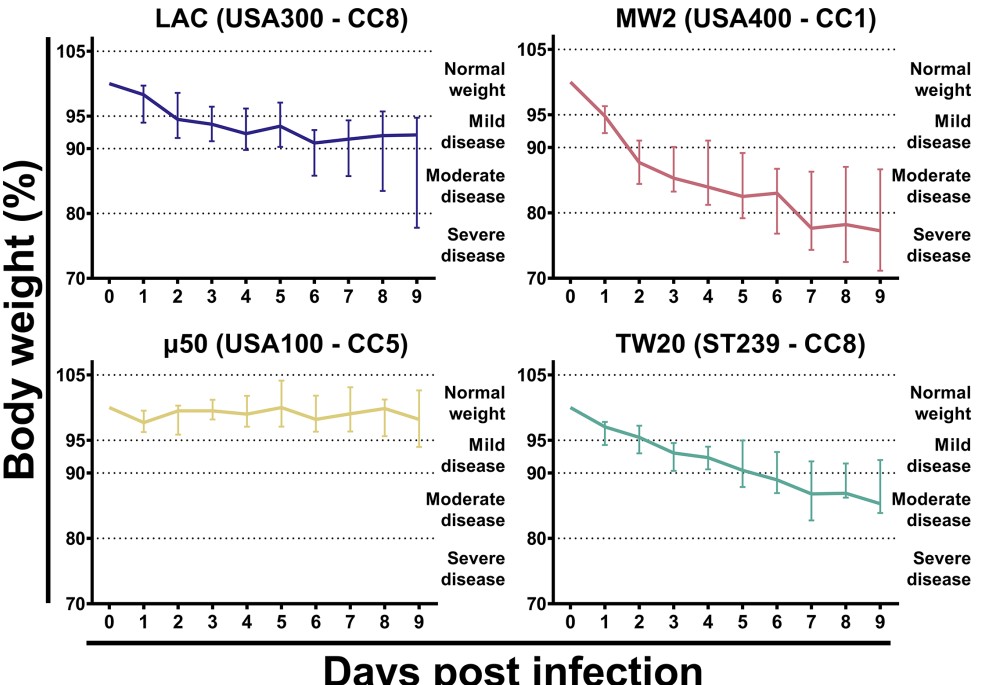

**FIG 1** Body weight loss of CD-1 mice after intravenous infection with different epidemiologically relevant *S. aureus* strains. Percentage of body weight loss in 8-week-old CD-1 female mice after intravenous infection with four *S. aureus* strains. Each graph shows the results obtained by infecting the animals with a single *S. aureus* strain. Data were normalized based on the body weight of each individual mouse just before infection. Solid lines represent the normalized median weight of the group of animals assessed at each time point, while the bars show the interquartile ranges (25% and 75% percentiles). Groups were represented by at least eight animals (range: 8–24 animals/group) from two independent experiments. Dashed lines define the frames for the severity of disease based on body weight loss.

maintained until the end of the experiment. In this case as well, the µ50 (USA100 – CC5) strain behaved differently from the other strains, as liver infection was almost cleared by 48 h after infection.

## The kinetics of *S. aureus* infection in inbred and outbred mice are very similar

Once the dynamics of infection in outbred mice (CD-1) had been evaluated, we studied what occurred in a commonly used inbred strain, C57BL/6N mice. Such an investigation was of particular interest, due to the widespread use of this mice strain in research, as results from these studies can inform subsequent experiments involving transgenic mice to explore specific immunological mechanisms. Therefore, C57BL/6N age-matched female animals were infected following the same protocol, and then bacterial load in blood, kidneys, and liver was characterized at 2 and 48 h p.i. Body weight loss served here as an indicator of general health conditions of mice. Sparingly, results were very similar to those obtained with outbred strains (Table 3). In detail, MW2 (USA400 – CC1) was confirmed to be the most virulent, considering both body weight loss and its ability to grow upon dissemination to the kidneys. By contrast, µ50 (USA100 – CC5) was again less able to disseminate (Fig. 3). Interestingly, *S. aureus* TW20 (ST239 – CC8) was able to reach higher concentrations in the blood at 1-hour p.i. compared to the other staphylococci in this mouse strain as well, even if the differences among bacterial strains were minimal (Table 4). In the kidneys, for all the strains, there was a 100-fold increase in bacterial counts from 2 to 48 h after infection, with some differences among groups at this latter time point (Fig. 3). In the liver, bacteria disseminated early, reaching high CFU counts ($1–3 \times 10^6$ CFU/organ) at 2 h p.i. and, again, at 48 h, the number decreased to roughly $10^3–10^4$ CFU/organ, except for mice infected with MW2 (USA100 – CC1) (Fig.

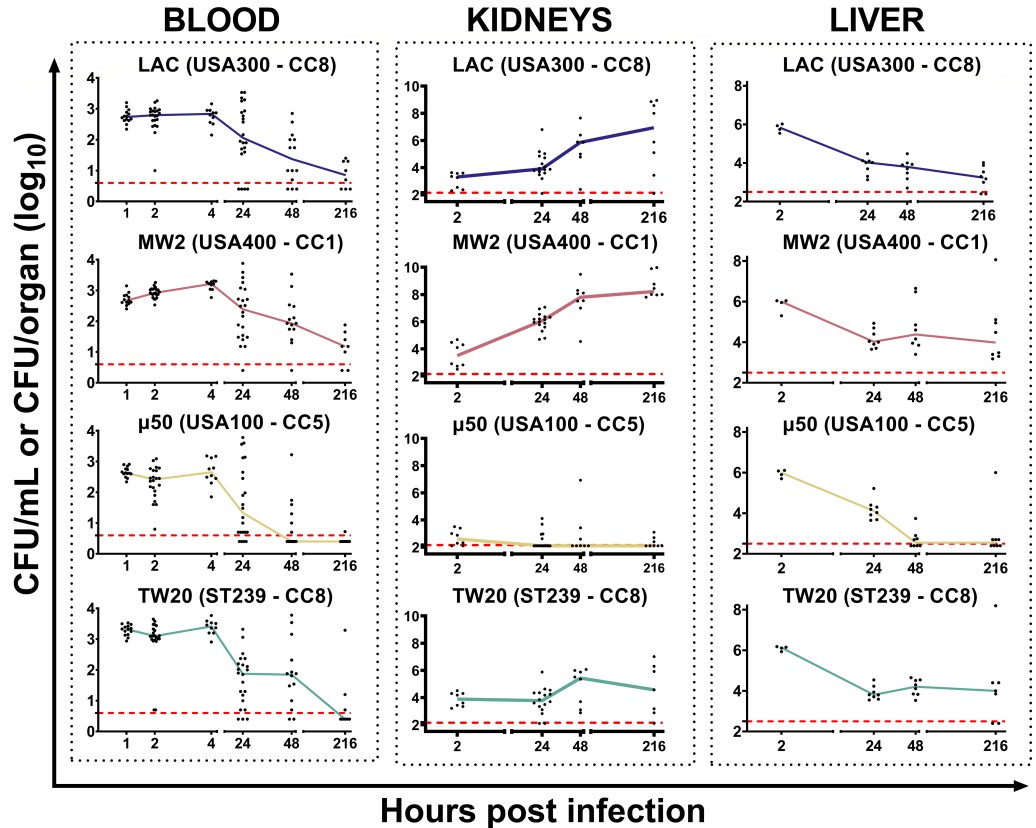

**FIG 2** CFU counts in the blood, kidneys, and liver following intravenous infection of CD-1 mice with different strains of *S. aureus*. CD-1 mice were intravenously infected with one of four *S. aureus* strains, and then bacterial concentration was assessed in the blood, kidneys, and liver of infected mice at the time points shown in the graphs. Each graph presents the results obtained by infecting the animals with a single *S. aureus* strain. Data were reported as log10 CFU counts per mL of blood or log10 CFU counts per organ (kidneys or liver). Each dot represents data from a single animal, and the line illustrates the trend of median values at each time point for individual groups. Data from 8 to 22 animals and at least two independent experiments were collected at each time point. In all the graphs, the red dotted line indicates the limit of detection of the assay.

3). Taken together, all these data suggest that the only significant difference observed between CD-1 outbred and C57BL/6N mice was the ability of MW2 (USA400 – CC1) to survive in the liver of C57BL/6N mice at significantly higher levels than in CD-1 (Fig. 3).

Understanding the kinetics of infection could help guide the design of new experiments, considering the characteristics of different bacterial and mouse strains that influence disease outcome through various host-pathogen interactions. This approach can be used to characterize other infection models to gain a comprehensive view of the interactions between *S. aureus* and the host in disease development, opening the possibility for novel approaches against infections.

## DISCUSSION

Multiple animal models of *S. aureus* infections were developed to improve our understanding of the disease processes and to assess the efficacy of new treatments. Among these, murine models of bacteremia and sepsis are widely used and characterized by bacterial counts and abscess formation in kidneys as primary endpoints (18–20). Despite being used for more than 30 years, they have been scarcely characterized, and some peculiar differences between human and mouse infections are not fully understood (18). In particular, the main difference lay in the kinetics of infection: while in humans, the spreading of bacteria to organs (sepsis) depends on bacterial growth in the blood (bacteremia), *S. aureus* is not able to efficiently persist in mouse blood, but infection can nevertheless progress to sepsis (19). Interestingly, mice, the formation of abscesses in

**TABLE 2** Summary of the comparisons among different strains in CD-1 mice[a,b]

| Organ/Blood | Time point (hours) | LAC[1] vs MW2[2] | LAC[1] vs μ50[3] | LAC[1] vs TW20[4] | MW2[2] vs μ50[3] | MW2[2] vs TW20[4] | μ50[3] vs TW20[4] |
|---|---|---|---|---|---|---|---|
| Blood | 1 | ns | ns | ***[4] | ns | ***[4] | ****[4] |
| | 2 | ns | *[1] | ***[4] | ***[2] | 0.06[4] | ****[4] |
| | 4 | *[2] | ns | ****[4] | **[2] | 0.06[4] | ****[4] |
| | 24 | ns | ns | ns | 0.06[2] | 0.07[2] | ns |
| | 48 | ns | 0.07[1] | ns | ***[2] | ns | **[4] |
| | 216 | ns | 0.06[1] | ns | **[2] | ns | ns |
| Kidneys | 2 | ns | ns | 0.07[4] | *[2] | ns | **[4] |
| | 24 | **[2] | **[1] | ns | ****[2] | ****[2] | *[4] |
| | 48 | 0.06[2] | *[1] | ns | ****[2] | *[2] | 0.08[4] |
| | 216 | ns | **[1] | ns | ****[2] | *[2] | ns |
| Liver | 2 | ns | ns | *[4] | ns | ns | ns |
| | 24 | ns | ns | ns | ns | ns | ns |
| | 48 | 0.07[2] | *[1] | ns | ****[2] | ns | ***[4] |
| | 216 | 0.08[2] | ns | ns | **[2] | ns | 0.05[4] |

[a]* = $P \leq 0.05$; ** = $P \leq 0.01$; *** = $P \leq 0.001$; **** = $P \leq 0.0001$; ns = not significant, $P > 0.05$. For $P \geq 0.05$ and $<0.1$, the value has been reported.

[b]To facilitate comprehension of the table, the numbers from 1 to 4 indicated in superscript indicate the *S. aureus* strains: 1) LAC (USA300 – CC8); 2) MW2 (USA400 – CC1); 3) μ50 (USA100 – CC5); 4) TW20 (ST239 – CC8). The numbers next to the asterisks or values in the table cells indicate the strain with the highest CFU counts for that sample in a given comparison.

organs and tissues is a prerogative of these infections, while in humans, even if common, this was not always the case (42).

In the present study, we characterized a model of bacteremia/sepsis in two different mouse strains using four epidemiologically relevant *S. aureus* strains with the aim of better understanding host-pathogen interactions. Based on our results: (i) independent of the bacterial strain used, except for μ50 (USA100 – CC5), *S. aureus* was able to survive in the blood for a long time period (at least 9 days), even if at a very low concentration; (ii) bacterial spreading to kidneys and liver followed vastly different pathways, confirming that kidneys are one of the main *S. aureus* target organs, while the liver is the dominant organ for cleaning bloodborne bacteria. (iii) No major differences among bacterial strains and between the selected outbred and inbred mouse strains were identified, underlining that the results obtained can be used to describe the host-pathogen interaction in this mouse model more broadly (41, 43–45).

Immediately after intravenous administration, *S. aureus* utilized the blood circulation to spread into the organs. While passing through the liver, most of the bacteria were captured (50%–80% of the inoculum) and subsequently killed (Fig. 2 and 3). Despite this, with the exception of a single low-virulent strain, the bacteria were still able to reach the organs, such as the kidneys and others, as reported in the literature, where they grew very fast over the following 2 days (31). Then, likely due to the formation of abscesses that limit bacterial growth *in situ*, a steady-state condition was achieved, and CFU numbers did not increase until the end of the observation. Meanwhile, the

**TABLE 3** Body weight loss of C57BL/6N mice after intravenous infection with different epidemiologically relevant *S. aureus* strains[a]

| Strain | Day 1 median weight (%) | Day 1 10–90 percentile (%) | Day 2 median weight (%) | Day 2 10–90 percentile (%) |
|---|---|---|---|---|
| LAC (USA300 – CC8) | 95.90 | 92.60–98.88 | 91.20 | 87.83–94.48 |
| MW2 (USA400 – CC1) | 90.55 | 86.809–3.58 | 85.10 | 81.88–89.05 |
| μ50 (USA100 – CC5) | 99.50 | 95.30–100.00 | 102.0 | 98.20–103.50 |
| TW20 (ST239 – CC8) | 94.30 | 90.75–95.75 | 90.35 | 86.88–93.20 |

[a]Data from three independent experiments and 12 mice per group were reported as the residual percentage of body weight compared with mouse weight before infection. Median and 10–90 percentile values at day 1 and day 2 after infection were reported for all the tested strains.

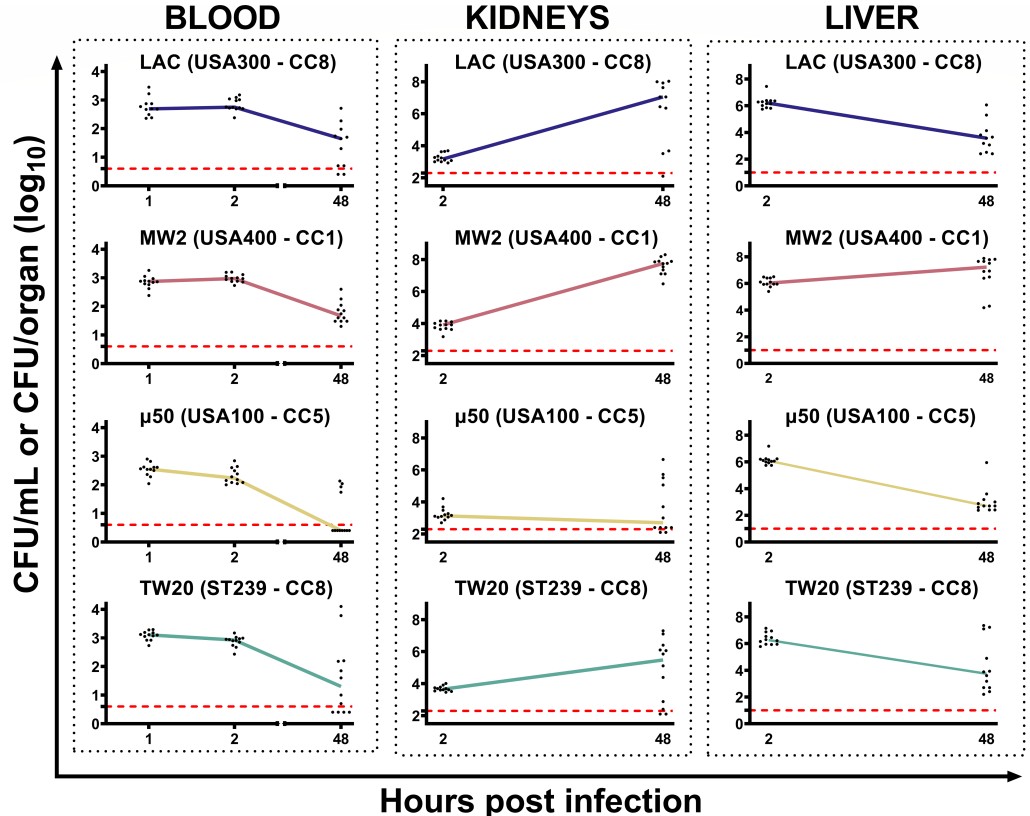

**FIG 3** CFU counts in the blood, kidneys, and liver following intravenous infection of C57BL/6N mice with different strains of *S. aureus*. C57BL/6N mice were intravenously infected with one of four *S. aureus* strains, and then bacterial concentration in the blood was assessed at the time points shown in the graphs. Each graph presents the results obtained by infecting the animals with a single *S. aureus* strain. Data were reported as log10 CFU counts per mL of blood or log10 CFU counts per organ (kidneys or liver). Each dot represents data from a single animal, and the line illustrates the trend of median values at each time point for individual groups. Data from 12 animals and three independent experiments were collected at each time point. In all the graphs, the red dotted line indicates the limit of detection of the assay.

number of bacteria in the blood decreased very fast in the first hours of infection and continued to decrease during the following days, even though a bacteremic state remained evident 9 days after infection for almost all the bacterial strains used (Fig. 2). We concluded that a first acute phase of infection persisted for a few days, during which bacteria spread to organs, grew very fast, and the animals showed a marked drop in body weight. Then, it appears that the host defense mechanisms likely contributed to controlling the infection, leading to a steady-state condition. The low-level persistence of bacteria in blood, observed until 9 days p.i. (Fig. 2), could be due to several mechanisms—for example, continuous exchange with bacteria entrapped in abscesses in the kidneys and other organs or bacteria surviving in host blood cells. Regarding this last point, it has been reported that multiple cell types can provide a protective niche for *S. aureus*, enabling the survival of persistent bacteria, undetectable by the immune system (46–48). In addition, *in vivo* selection of particularly resistant spontaneous mutants can be an explanation of this unexpected phenomenon (49, 50). Even though no major differences among *S. aureus* strains were identified, some minor peculiarities were found that correlated with bloodstream infection in humans. For instance, the acquisition of TW20 (ST239 – CC8) was found to be fourfold more likely associated with bacteremia in humans than the acquisition of other epidemiologically relevant MRSA strains (29). Here, we showed that TW20 (ST239 – CC8) exhibited an evident tropism for blood compared with the other strains (Tables 2 and 4). To our knowledge, this study is the first to report the prolonged survival of *S. aureus* in mouse blood, with notable differences

**TABLE 4** Summary of the comparisons among different strains in C57BL/6N mice[a,b]

| Organ/blood | Time point (hours) | LAC[1] vs MW2[2] | LAC[1] vs µ50[3] | LAC[1] vs TW20[4] | MW2[2] vs µ50[3] | MW2[2] vs TW20[4] | µ50[3] vs TW20[4] |
|---|---|---|---|---|---|---|---|
| Blood | 1 | ns | ns | **[4] | **[2] | *[4] | ****[4] |
| | 2 | ns | ***[1] | ns | ****[2] | ns | ***[4] |
| | 48 | ns | ns | ns | *[2] | ns | ns |
| Kidneys | 2 | ***[2] | ns | **[4] | ***[2] | ns | **[4] |
| | 48 | 0.08[2] | **[1] | 0.09[1] | ****[2] | ***[2] | ns |
| Liver | 2 | ns | ns | ns | ns | ns | ns |
| | 48 | ***[2] | ns | ns | ****[2] | **[2] | 0.09[4] |

[a]* = $P \leq 0.05$; ** = $P \leq 0.01$; *** = $P \leq 0.001$; **** = $P \leq 0.0001$; ns = not significant, $P > 0.05$. For $P \geq 0.05$ and $< 0.1$, the value has been reported.
[b]To facilitate comprehension of the table, the numbers from 1 to 4 were associated with each single *S. aureus* strain. List of strains with the related identification numbers: 1) LAC (USA300 – CC8); 2) MW2 (USA400 – CC1); 3) µ50 (USA100 – CC5); 4) TW20 (ST239 – CC8). The numbers next to the asterisks or values in the table cells indicate the strain with the highest CFU counts for that sample in a given comparison.

among strains, suggesting potential use for a more reliable murine model of bacteremia. During the initial phase of infection, most of the bacteria were entrapped and killed in the liver, likely by Kupffer cells (Fig. 2 and 3). It has been described that encapsulated bacteria are susceptible to hepatic capture and killing, and since many *S. aureus* strains have undergone capsule loss to alleviate metabolic costs, this could explain why *S. aureus* was so efficiently cleared by the liver (43, 51–53). This organ is, therefore, the first line of defense against *S. aureus* infections. Translating our findings to humans, our data confirm that the health of the liver is essential in preventing primary bacteremic infections from becoming septic infections (54). As a matter of fact, the liver seemed to reach a state where it was unable to efficiently remove all the bacteria, possibly because it required time to recover from *S. aureus*-triggered damage. Therefore, bacteria were not completely cleared, and they survived in the liver and in the blood for at least 9 days after infection (Fig. 2 and 3). Interestingly, the only significant difference between outbred and inbred mice was the ability of MW2 (USA400 – CC1) to survive in the liver of C57BL/6N mice at higher levels than in CD-1 mice.

It appeared that the ability of this mouse strain to kill entrapped bacteria was, for some reason, reduced. Why this occurred is still unclear and needs additional studies for clarification, but some possible explanations can be considered. For example, the MW2 (USA400 – CC1) strain was shown to be the most aggressive among the tested strains due to its ability to grow faster in the kidneys (Fig. 2 and 3). As a consequence, this resulted in an enhanced ability to also survive in the liver, which was particularly evident in C57BL/6N mice (Fig. 3). Notably, C57BL/6N mice are defined as Th1-prone, and this could influence cell-mediated immunity by affecting the balance of M1/M2 Kupffer cell activation (55). Indeed, M1 Kupffer cells are pro-inflammatory, releasing cytokines like IL-12 and IL-23, which can contribute to the defense against pathogens but at the same time trigger inflammation and liver damage. In contrast, M2 Kupffer cells are anti-inflammatory, releasing molecules like IL-10 and TGF-β, and promoting tissue repair (56). It has been reported that C57BL/6N mice naturally possess a lower proportion of M2 Kupffer cells compared to other mouse strains, demonstrating lower resistance to alcohol-induced liver damage (57, 58). This highlights the potential protective role of M2 Kupffer cells against other mechanisms that can damage the liver.

One of the tested strains, µ50 (USA100 – CC5), was less pathogenic compared withthe others in mice. Strains of the *S. aureus* USA100 clone are the predominant colonizers of human nares in the United States and a leading cause of invasive infection, primarily among persons with healthcare-associated risk factors (59, 60). Therefore, this difference could be due to the relatively low dose used to infect the animals or the fact that mice do not present specific underlying conditions for the strain to disseminate and cause infections.

In contrast, MW2 (USA400 – CC1) was more pathogenic compared with the others in mice. Interestingly, in humans, this strain is responsible for several cases of aggressive

MRSA infections that occur among individuals in the United States without established risk factors and is associated with severe infections like necrotizing pneumonia, pulmonary abscesses, and sepsis (61).

In a possible follow-up study, it would make sense to extend the observation period to follow the infection kinetics over longer time points. As described in the literature, for example, staphylococci may be released from mature abscesses into circulation, initiating new rounds of infection (19). Characterizing this aspect for several *S. aureus* and mouse strains could be beneficial in identifying new possible windows for preventive/therapeutic interventions that more closely mimic human conditions.

In conclusion, this approach revealed the presence of persisting *S. aureus* in the blood that can sustain bacteremia. Additionally, this model highlighted the role of the liver in the clearance of *S. aureus* from the blood, a role that could be exploited against bloodstream infections. Our study reaffirms established principles of *S. aureus* pathogenesis while identifying new directions for research into how bacterial strain variation and host genetics influence disease progression and treatment response. Although common mechanisms underlying bacteremia were evident across the different strains used, subtle strain-specific differences suggest that some peculiarities can modulate infection dynamics. These results underscore the importance of investigating strain-dependent infectivity and pathogenesis to possibly identify precise molecular targets responsible for a disease and to refine preventive/therapeutic strategies that account for *S. aureus* strain variation.

## ACKNOWLEDGMENTS

A special acknowledgment is due to all the GSK Animal Resources Center, Siena, Italy, for their excellent support with the *in vivo* studies. During the preparation of this work, a GSK proprietary large language model was used to assist with editing the manuscript. After using this tool, the authors reviewed and edited the content as needed and took full responsibility for the content of the published article.

S.V.: Data curation, Methodology, Writing – original draft, Writing – review & editing. M.T.: Methodology, Writing – review & editing. S.T.: Methodology, Writing – review & editing. M.B.: Writing – review & editing, Supervision. M.M.D.: Formal analysis, Supervision, Writing – review & editing E.C.: Data curation, Formal analysis, Methodology, Supervision, Writing – original draft, Writing – review & editing. All authors reviewed and approved the final version of the manuscript.

## AUTHOR AFFILIATIONS

[1]GSK, Siena, Italy

[2]Department of Biology, University of Rome Tor Vergata, Rome, Italy

[3]PhD Program in Evolutionary Biology and Ecology, Department of Biology, University of Rome "Tor Vergata", Rome, Italy

## AUTHOR ORCIDs

Emiliano Chiarot  http://orcid.org/0000-0002-5574-116X

## AUTHOR CONTRIBUTIONS

Serena Vastola, Conceptualization, Data curation, Formal analysis, Writing – original draft, Writing – review and editing | Marco Tortoli, Methodology | Stefania Torricelli, Methodology | Michela Brazzoli, Conceptualization, Supervision, Writing – original draft | Marco Maria D'Andrea, Supervision, Writing – original draft, Writing – review and editing | Emiliano Chiarot, Conceptualization, Formal analysis, Supervision, Writing – original draft, Writing – review and editing

## DATA AVAILABILITY

The original contributions presented in the study are included in the article or Supplemental material. Further inquiries can be directed to the corresponding author.

## ETHICS APPROVAL

The animal study was approved by the Italian Ministry of Health (authorization n° 520/2023-PR). The study was conducted in accordance with local legislation and institutional requirements. GSK is committed to the replacement, reduction, and refinement of animal studies (3Rs). When animals are required, application of robust study design principles and peer review minimizes animal use, reduces harm, and improves benefit in studies.

## ADDITIONAL FILES

The following material is available online.

### Supplemental Material

**Supplemental figures and tables (Spectrum02642-25-s0001.docx).** Figure S1 and Tables S1 to S6.

### Open Peer Review

**PEER REVIEW HISTORY (review-history.pdf).** An accounting of the reviewer comments and feedback.

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
