## [Reviewer comments · Microbiology Spectrum]

Microbiology Spectrum

Unravelling the mouse model of *Staphylococcus aureus* bacteraemia and sepsis: a systematic approach to better characterize host/pathogen interactions

Serena Vastola, Marco Tortoli, Stefania Torricelli, Michela Brazzoli, Marco D'Andrea, and Emiliano chiarot

Corresponding Author(s): Emiliano chiarot, GSK Vaccines SRL

Review Timeline:

Submission Date:	August 26, 2025
Editorial Decision:	October 3, 2025
Revision Received:	December 9, 2025
Accepted:	December 12, 2025

Editor: M. Victoria Delpino

Reviewer(s): Disclosure of reviewer identity is with reference to reviewer comments included in decision letter(s). The following individuals involved in review of your submission have agreed to reveal their identity: Basel H Abuaita (Reviewer #1); sreekanth reddy basireddy (Reviewer #2)

Transaction Report:

DOI: <https://doi.org/10.1128/spectrum.02642-25>

Re: Spectrum02642-25 (**Unravelling the mouse model of *Staphylococcus aureus* bacteraemia and sepsis: a systematic approach to better characterize host/pathogen interactions**)

Dear Dr. Emiliano chiarot:

Thank you for the privilege of reviewing your work. Below you will find my comments, instructions from the Spectrum editorial office, and the reviewer comments.

Following peer review, please prepare and submit a formal response to the reviewers' suggestions and a revised manuscript addressing their comments.

Revision Guidelines

Sincerely,
M. Victoria Delpino
Editor
Microbiology Spectrum

Reviewer #1 (Comments for the Author):

The manuscript by Vastola et al aims to better characterize the mouse model of *Staphylococcus aureus* bacteremia/sepsis using multiple different bacterial isolates and two different mouse strains (CD-1 and C57BL/6N). Overall, providing a course time assessment of bacterial burdens in various tissues and psychological symptoms of mice in response to *Staphylococcus aureus*

systemic infections is valuable information for the field of *Staphylococcus aureus* pathogenesis and host responses. However, the manuscript is poorly written, and I have some concerns that need to be addressed. I recommend having a native English language colleague proofread the manuscript. Also, several studies have shown that infections with USA300 with similar inoculation sizes cause lethality in mice (PMID: 23555002, PMID: 30721149, PMID: 36681080, PMID: 34871043). Thus, the authors must provide mouse survival plots for all their experiments. The following are my other concerns:

Concerns:

1. A well-written paragraph should begin with a topic sentence. Several paragraphs don't have a topic sentence. Also, it is better not to start multiple paragraphs and/or sentences with "at". Several extra spaces should be removed. Examples are extra spaces located between lines 22-23, 32-33, 102-103, 111-112, 124-125, 143-144, 150-151, 175-176, and 182-183.
2. For the statement in lines 72-73, the authors should provide the actual reference instead of a review since it has a specific emphasis on *Staphylococcus aureus* bacterial rapid growth once it reaches the blood in humans, which is different in mice.
3. It is not clear why the authors selected CD-1 and C57BL/6N mouse strains. Most of the studies in the field of *Staphylococcus aureus* pathogenesis use either BALB/c as a susceptible mouse strain or C57BL/6J as a more resistant strain. The C57BL/6J is also used because of the availability of various gene knockouts. The authors must provide data for these strains since they are the most common mouse strains used in the field.
4. In line 99, the authors claim the bacteria at an OD₆₀₀ of 2 are in the early exponential phase. This is incorrect. The bacteria are in the late exponential phase. If this is not a typo, the authors must compare the infectivity between bacteria grown at early and late exponential phases. There will be dead bacteria present at the late exponential phase, which could influence the inoculum size.
5. In lines 144-146, it states that blood was collected from mice at a final concentration of 500 UI/ml or above. How much is that? The authors must provide the volume of the blood instead.
6. Table 1 legend indicates that the table provides CFU counts in blood, kidneys, and liver. However, the table only provides statistical analysis for Figure 2. The authors should combine Figure 2 and Table 1 into one figure to better indicate the statistical significance of the data in the figure.
7. Table 3 legend also provides statistical analysis for Figure 3 and not actual CFU counts. The authors should combine the Table with Figure 3.
8. The conclusion in lines 186-189 contradicts the statement in lines 189-191. Based on the results of Figure 1, the authors concluded that all mice lost weight upon infection, but they contradict this conclusion by stating that USA100 infection did not cause loss of weight. The authors must correct the contradictory statements.
9. In lines 214-217, the authors conclude that the TW20 strain persists better in the blood during the early phase of infection when compared to other strains. The "persist" should be changed to "survive" since persistence is referred to as survival at the late stage of infection. Now, since USA300 and USA400 survived longer at day 9, this may indicate that these strains persist better than the others in the blood.
10. Lastly, I recommend that the authors provide a summary at the end of the results section where they can explain how the new findings should guide future research in assessing the infectivity or the pathogenesis of different strains of *Staphylococcus aureus*.

Reviewer #2 (Comments for the Author):

The authors present a well-designed and timely study that characterizes a murine intravenous infection model of *Staphylococcus aureus* bacteraemia and sepsis using four epidemiologically important MRSA lineages in both outbred CD-1 and inbred C57BL/6N mice, detailing bacterial kinetics in blood, liver, and kidneys to reveal three infection phases including rapid hepatic clearance, kidney dissemination with abscess formation, and prolonged low-level bacteraemia. Strengths of the work include a clear rationale that fills a long-standing knowledge gap, robust experimental design with randomization and blinding, inclusion of multiple clinically relevant strains and two genetic backgrounds, and the novel observation of persistent low-level bacteraemia with detailed organ tropism, all of which provide an excellent reference for therapeutic and vaccine research. Weaknesses and limitations are that the study remains largely descriptive without mechanistic immune analysis, follows the inbred C57BL/6N mice only to 48 h despite an initial implication of longer monitoring, and uses a single intravenous dose and route without dose-response assessment.

However, some points need clarification and minor revision:

1. Lines 121-126: There appears to be an inconsistency regarding the observation period for C57BL/6N mice, as the methods section mentions monitoring for up to 9 days while the Results and Supplementary Figure S1 describe sampling at only 2 hours and 48 hours, and it would be helpful if the authors could briefly clarify this aspect to ensure consistency and avoid any potential confusion for readers.
2. Line 21-22: Rephrase increasing predictivity as compared to human, for clarity.
3. Line 28: Use signs of disease instead of symptoms when referring to animals.
4. Table 2: Correct the typo 95-75 to 95.75 for TW20

Reviewer #3 (Comments for the Author):

This manuscript studies a murine model for bacteremia and sepsis caused by *Staphylococcus aureus*.
The following comments are made:

1. Lines 15-16, 51. Currently, evidence also points to the pharynx as an important colonization niche in humans. Review and include the evidence in this regard.
2. Line 151. Explain how euthanasia was performed.
3. Line 155. Were no dilutions performed for the microbial count? Explain.
4. Lines 176-185. This information can be included in the introduction or in Methods, leaving only the results for the Results section.
5. Line 216. Add only the letter p. Correct the entire text and tables.
6. Lines 302-303. To which part of the immune response are these referring? Explain.

The authors present a well-designed and timely study that characterizes a murine intravenous infection model of *Staphylococcus aureus* bacteraemia and sepsis using four epidemiologically important MRSA lineages in both outbred CD-1 and inbred C57BL/6N mice, detailing bacterial kinetics in blood, liver, and kidneys to reveal three infection phases including rapid hepatic clearance, kidney dissemination with abscess formation, and prolonged low-level bacteraemia. Strengths of the work include a clear rationale that fills a long-standing knowledge gap, robust experimental design with randomization and blinding, inclusion of multiple clinically relevant strains and two genetic backgrounds, and the novel observation of persistent low-level bacteraemia with detailed organ tropism, all of which provide an excellent reference for therapeutic and vaccine research. Weaknesses and limitations are that the study remains largely descriptive without mechanistic immune analysis, follows the inbred C57BL/6N mice only to 48 h despite an initial implication of longer monitoring, and uses a single intravenous dose and route without dose-response assessment.

However, some points need clarification and minor revision:

1. Lines 121–126: There appears to be an inconsistency regarding the observation period for C57BL/6N mice, as the methods section mentions monitoring for up to 9 days while the Results and Supplementary Figure S1 describe sampling at only 2 hours and 48 hours, and it would be helpful if the authors could briefly clarify this aspect to ensure consistency and avoid any potential confusion for readers.
2. Line 21-22: Rephrase increasing predictivity as compared to human, for clarity.
3. Line 28: Use signs of disease instead of symptoms when referring to animals.
4. Table 2: Correct the typo 95-75 to 95.75 for TW20

This manuscript studies a murine model for bacteremia and sepsis caused by *Staphylococcus aureus*.

The following comments are made:

1. Lines 15-16, 51. Currently, evidence also points to the pharynx as an important colonization niche in humans. Review and include the evidence in this regard.
2. Line 151. Explain how euthanasia was performed.
3. Line 155. Were no dilutions performed for the microbial count? Explain.
4. Lines 176-185. This information can be included in the introduction or in Methods, leaving only the results for the Results section.
5. Line 216. Add only the letter p. Correct the entire text and tables.
6. Lines 302-303. To which part of the immune response are these referring? Explain.

Unravelling the mouse model of *Staphylococcus aureus* bacteraemia and sepsis: a systematic approach to better characterize host/pathogen interactions

Serena Vastola^{1,2,3}, Marco Tortoli¹, Stefania Torricelli¹, Michela Brazzoli¹, Marco Maria D'Andrea², Emiliano Chiarot^{1*}

Affiliations

¹ GSK, Siena, Italy

² Department of Biology, University of Rome Tor Vergata, Rome, Italy

³ PhD Program in Evolutionary Biology and Ecology, Department of Biology, University of Rome "Tor Vergata", Rome, Italy

*Corresponding author

Keywords: murine model, bloodstream infection, MRSA, pathogenesis

To the Editorial Team and Reviewers,

We thank you for your thoughtful comments and for the opportunity to revise our manuscript. We appreciate the reviewers' suggestions, which have helped to improve the manuscript's clarity, integration with the current literature, and overall quality. Below, we provide point-by-point responses to each comment and detail the revisions made.

We believe we have addressed all concerns and incorporated the suggested changes. If any further modifications are required, please let us know and we will respond promptly.

Sincerely, Emiliano Chiarot on behalf of all authors.

Reviewer #1 (Comments for the Author):

The manuscript by Vastola et al aims to better characterize the mouse model of *Staphylococcus aureus* bacteremia/sepsis using multiple different bacterial isolates and two different mouse strains (CD-1 and C57BL/6N). Overall, providing a course time assessment of bacterial burdens in various tissues and psychological symptoms of mice in response to *Staphylococcus aureus* systemic infections is valuable information for the field of *Staphylococcus aureus* pathogenesis and host responses. However, the manuscript is poorly written, and I have some concerns that need to be addressed.

1. I recommend having a native English language colleague proofread the manuscript.

Thank you. We completed a second round of proofreading with a native English-speaking colleague and internal reviewers, and we believe the manuscript has now reached an acceptable level of quality.

2. Also, several studies have shown that infections with USA300 with similar inoculation sizes cause lethality in mice (PMID: 23555002, PMID: 30721149, PMID: 36681080, PMID: 34871043). Thus, the authors must provide mouse survival plots for all their experiments.

We did not provide any survival curve because all infected animals, regardless of S. aureus strain or mouse strain, survived the 9-day observation period after infection. To be honest, one mouse was found dead shortly after infection with the TW20 (ST239) strain; we attributed this to a probable injection problem rather than to the infection. Below we provide a comprehensive overview of the papers cited by the reviewer. In those publications, the authors used different S. aureus USA300 clones and/or mouse strains and/or infectious doses and/or routes of administration, so a direct comparison with our data is not possible. Finally, based on our experience with tens of S. aureus strains in this infection model, using an infectious dose close to 1.0×10^7 CFU per mouse administered by tail- vein injection in both young adult CD- 1 and C57BL/6N mice, mortality within 4–14 days after infection is very rare.

S. aureus strain	Mice strain, gender, age	Infection route	Dose (CFU/mouse)	Publication
NRS384 (USA300)	BALB/c, gender and age not specified	Tail vein injection	2×10^7	PMID: 23555002
USA300 not specified	C57BL/6J, female, 6-8 weeks old	Retro-orbital injection	5×10^7	PMID: 30721149
I1-I5, H1–5, H2 (USA300)	ND4 Swiss Webster, female, 5 weeks old	Retro-orbital injection	5×10^7 or 7.5×10^7	PMID: 36681080
LAC (USA300)	MICU ^{fl/fl} on the C57BL/6J background, female, 10 to 13 week-old	Retro-orbital injection	2×10^7	PMID: 34871043

The following are my other concerns:

Concerns:

3a. A well-written paragraph should begin with a topic sentence. Several paragraphs don't have a topic sentence.

We do not understand very well this point. All the paragraphs of the Results start with a brief introduction either linking it to the previous paragraph or explaining the main objective of the data presented in that specific paragraph. Here a couple of examples:

“We evaluated the ability of the selected strains to infect animals after intravenous administration with outbred mice chosen for the first set of experiments to better mimic the genetic diversity of the human population.”

*“Previous data showed that intravenous infection with *S. aureus* can result in a very short initial phase in which bacteria remain in the blood (bacteraemia phase) and a second phase (sepsis phase) when bacteria reach organs and tissues and form abscesses. To better characterize these two phases, we enumerated bacteria in the blood of infected animals...”*

Could you please clarify what you mean and, if necessary, suggest some improvements?

Thank you.

3b. Also, it is better not to start multiple paragraphs and/or sentences with "at". Several extra spaces should be removed. Examples are extra spaces located between lines 22-23, 32-33, 102-103, 111-112, 124-125, 143-144, 150-151, 175-176, and 182-183.

Thank you for these helpful points. We have indeed found 2 sentences in the methods section which started with “at”. Sentences have been changed following the suggestion. Lines 109-110 and 146-147 in the Marked-Up file.

Additionally, we found some extra spaces and removed them from the text as suggested.

We hope we have removed them all

4. For the statement in lines 72-73, the authors should provide the actual reference instead of a review since it has a specific emphasis on *Staphylococcus aureus* bacterial rapid growth once it reaches the blood in humans, which is different in mice.

We have now included primary references that specifically demonstrate the rapid growth of Staphylococcus aureus in human blood; the previously cited review has been removed. For your convenience, the list of citations we added is as follows:

- Khatib R. et al., 2005. Time to positivity in Staphylococcus aureus bacteremia: possible correlation with the source and outcome of infection. Clin Infect Dis 41:594-8.

- Minejima E. et al., 2020. Defining the Breakpoint Duration of Staphylococcus aureus Bacteremia Predictive of Poor Outcomes. Clin Infect Dis 70:566-573.

5. It is not clear why the authors selected CD-1 and C57BL/6N mouse strains. Most of the studies in the field of *Staphylococcus aureus* pathogenesis use either BALB/c as a susceptible mouse strain or C57BL/6J as a more resistant strain. The C57BL/6J is also used because of the availability of various gene knockouts. The authors must provide data for these strains since they are the most common mouse strains used in the field.

Thank you for the comment. The reasons we selected these strains are as follows.

*CD-1: An outbred strain that represents the natural variability of the human population. We have extensive experience with this mouse strain infected with many *S. aureus* isolates and have published studies using this disease model (DOI: 10.1128/IAI.00270-17; DOI: 10.1038/srep38043). We believe it is a good choice when outcome variability is an important parameter to assess.*

*C57BL/6N: An inbred strain that is less susceptible to infection than BALB/c, another inbred strain frequently used in *S. aureus* research. We selected this Th1-biased strain because it tends toward Th1/Th17-type responses, which are more closely aligned with effective anti-*S. aureus* immunity (for example, IFN- γ , IL-17, and strong neutrophil responses). There is a*

large literature using this strain in S. aureus bacteremia models; in many cases the J vs. N genetic background is not specified.

We acknowledge that our data provide only a partial view of the field, as additional experiments and information would be required for a more complete understanding. Nevertheless, we believe these results can positively contribute to understanding host–pathogen dynamics in S. aureus bacteremia models.

6. In line 99, the authors claim the bacteria at an OD600 of 2 are in the early exponential phase. This is incorrect. The bacteria are in the late exponential phase. If this is not a typo, the authors must compare the infectivity between bacteria grown at early and late exponential phases. There will be dead bacteria present at the late exponential phase, which could influence the inoculum size.

We would like to clarify that this is not a typographic error. We performed a detailed growth curve analysis for this bacterium under our experimental conditions. Bacteria were grown in Tryptic Soy Broth (TSB) starting from a frozen aliquot. The cultures were grown in a 50 mL volume at an OD600nm of 0.05, using 250 mL disposable Erlenmeyer flasks with 0.2 µm ventilated caps. The bacteria were incubated at 37°C with agitation at 250 rpm. Our spectrophotometer exhibits linear behavior at OD600nm between values of 0.05 and 1.1. To ensure accuracy, we performed appropriate dilutions when measuring samples with OD600nm values outside of this range. This procedure guarantees that all readings, including the reported value of 2, accurately reflect the bacterial growth phase. Our data indicates that the late exponential phase for all the strains evaluated begins at OD600nm of approximately 10-12. A bacterial growth with optical density of 2.0 at 600nm is in the early exponential phase and corresponds to 0.5-1.5 X 10⁹ CFU/mL. Based on these observations, we are confident in our characterization of the bacterial growth phase, and we believe that differences in infectivity associated with growth phase do not need further evaluation under these conditions.

7. In lines 144-146, it states that blood was collected from mice at a final concentration of 500 UI/ml or above. How much is that? The authors must provide the volume of the blood instead.

Thank you for your comment, we have noticed a typo. The blood volume collected from mice was not fixed and varied between approximately 50 and 250 μ L depending on mouse strain, mouse weight, number of bleedings and technical conditions during collection. However, we ensured that sufficient heparin was added to achieve or exceed the final concentration of 200 UI/mL in every sample. As reported in literature (PMID: 21490828) we have used a heparin concentration that reliably guarantees its efficacy as an anticoagulant agent.

8. Table 1 legend indicates that the table provides CFU counts in blood, kidneys, and liver. However, the table only provides statistical analysis for Figure 2. The authors should combine Figure 2 and Table 1 into one figure to better indicate the statistical significance of the data in the figure.

We have changed the title of Table 1 from "Table 1: Summary of the comparisons among different strains in CD-1 mice: CFU counts in blood, kidneys, and liver" to "Table 1: Summary of the comparisons among different strains in CD-1 mice," removing the reference to CFU counts to make the title clearer. We prefer not to combine Table 1 and Figure 2 so as to facilitate comprehension of the message we wish to convey. In our view, it would be very difficult to graphically display all the comparisons made in a single figure without causing confusion.

These modifications have been done in lines 444-445 of the "Marked-Up" document.

9. Table 3 legend also provides statistical analysis for Figure 3 and not actual CFU counts. The authors should combine the Table with Figure 3.

We have changed the title of Table 3 as reported: from "Table 3: Summary of the comparisons among different strains in C57BL/6N mice: CFU counts in blood, kidneys, and liver" to "Table 3: Summary of the comparisons among different strains in C57BL/6N mice"

removing the reference to CFU counts to make the clearer. Also here, we prefer not to combine Table 3 and Figure 3 to facilitate the comprehension of the message we would like to deliver.

These modifications have been done in lines 460-461 of the "Marked-Up" document.

10. The conclusion in lines 186-189 contradicts the statement in lines 189-191. Based on the results of Figure 1, the authors concluded that all mice lost weight upon infection, but they contradict this conclusion by stating that USA100 infection did not cause loss of weight. The authors must correct the contradictory statements.

We have changed the statement and made corrections that, we believe, fit the reviewer's comment. In line 181, the statement "In general, we concluded that almost all the infected mice showed a decrease in body weight during the first 4–7 days after infection..." has been modified to "In general, we concluded that almost all the mice infected with 3 out of 4 strains showed a decrease in body weight during the first 4–7 days after infection..."

11. In lines 214-217, the authors conclude that the TW20 strain persists better in the blood during the early phase of infection when compared to other strains. The "persist" should be changed to "survive" since persistence is referred to as survival at the late stage of infection. Now, since USA300 and USA400 survived longer at day 9, this may indicate that these strains persist better than the others in the blood.

*We agree that the term "survive" more accurately describes the early-phase dynamics observed in our data, whereas "persistence" should be reserved for later-stage observations. Accordingly, we have revised the text to replace "persist" with "survive" to better reflect our observations. The sentence in line 211 of the Marked-Up file is now: "CFU counts in the blood of animals infected with *S. aureus* TW20 (ST239 – CC8) were 3-fold higher compared to the other strains ($p < 0.01$), suggesting that this strain could survive better in the blood (Fig. 2 and Table 1).*

12. Lastly, I recommend that the authors provide a summary at the end of the results section where they can explain how the new findings should guide future research in assessing the infectivity or the pathogenesis of different strains of *Staphylococcus aureus*.

We appreciate the reviewer's suggestion. We moved the more technical statement "Understanding the kinetics of infection could help guide the design of new experiments, considering the characteristics of different bacterial and mouse strains that influence disease outcome through various host-pathogen interactions. This approach can be used to characterize other infection models to gain a comprehensive view of the interactions between S. aureus and the host in disease development, opening the possibility for novel approaches against infections" from the Discussion into the Results (now lines 260-265 of the Marked-Up document) and replaced it with a succinct conclusion that better situates our work within the field. This summary is reported in lines 370-377 in the Marked-Up document. "Our study reaffirms established principles of S. aureus pathogenesis while identifying new directions for research into how bacterial strain variation and host genetics influence disease progression and treatment response. Although common mechanisms underlying bacteraemia were evident across the different strains used, subtle strain-specific differences suggest that some peculiarities can modulate infection dynamics. These results underscore the importance of investigating strain-dependent infectivity and pathogenesis to possibly identify precise molecular targets responsible for a disease and to refine preventive/therapeutic strategies that account for S. aureus strain variation."

Reviewer #2 (Comments for the Author):

The authors present a well-designed and timely study that characterizes a murine intravenous infection model of *Staphylococcus aureus* bacteraemia and sepsis using four epidemiologically important MRSA lineages in both outbred CD-1 and inbred C57BL/6N mice, detailing bacterial kinetics in blood, liver, and kidneys to reveal three infection phases including rapid hepatic clearance, kidney dissemination with abscess formation, and prolonged low-level bacteraemia. Strengths of the work include a clear rationale that fills a long-standing knowledge gap, robust experimental design with randomization and blinding, inclusion of multiple clinically relevant strains and two genetic backgrounds, and the novel observation of persistent low-level bacteraemia with detailed organ tropism, all of which provide an excellent reference for therapeutic and vaccine research. Weaknesses and limitations are that the study remains largely descriptive without mechanistic immune analysis, follows the inbred C57BL/6N mice only to 48 h despite an initial implication of longer monitoring, and uses a single intravenous dose and route without dose-response assessment.

However, some points need clarification and minor revision:

1. Lines 121-126: There appears to be an inconsistency regarding the observation period for C57BL/6N mice, as the methods section mentions monitoring for up to 9 days while the Results and Supplementary Figure S1 describe sampling at only 2 hours and 48 hours, and it would be helpful if the authors could briefly clarify this aspect to ensure consistency and avoid any potential confusion for readers.

Thank you for the opportunity to clarify. When we referred to “up to 9 days,” we meant that 9 days represents the maximum observation period. Not all mice were, nevertheless, observed for the full duration. Specifically, as the reviewer specified well, CD-1 mice were monitored for all this period, while C57BL/6N mice, only for 2 days. Since no major differences

occurred between observations done at day 2 and at day 9 in CD-1 mice, we stopped observations at the earlier timepoint for C57BL/6N mice for ethical reasons.

To be clearer, we modified the Methods (lines 119-120 in the Marked-Up document) as follows: "Mice were observed daily for the development of clinical signs of disease and body weight loss for up to 9 days (CD-1 mice) or 2 days (C57BL/6N mice) after infection."

2. Line 21-22: Rephrase increasing predictivity as compared to human, for clarity.

Thank you for the comment. We rephrased it as follows "while better reflecting human disease dynamics", lines 20-21 in the Marked-Up file.

3. Line 28: Use signs of disease instead of symptoms when referring to animals.

Thanks, we agree that the use of "signs of disease" instead of "symptoms" is more appropriate. We have made changes accordingly, in particular line 26 and line 137 in the Marked-Up file.

4. Table 2: Correct the typo 95-75 to 95.75 for TW20

Thank you for pointing this error out. We have corrected the typographical error in Table 2 accordingly.

Reviewer #3 (Comments for the Author):

This manuscript studies a murine model for bacteremia and sepsis caused by *Staphylococcus aureus*. The following comments are made:

1. Lines 15-16, 51. Currently, evidence also points to the pharynx as an important colonization niche in humans. Review and include the evidence in this regard.

Thank you, a citation has been added, and the statements have been changed accordingly as follows: "Staphylococcus aureus is a pathobiont whose primary human reservoirs are nares, pharynx, intestines, and skin", lines 14-15 of the Marked-Up document.

2. Line 151. Explain how euthanasia was performed.

To clarify, euthanasia was performed by cervical dislocation. Following the procedure, clinical signs were carefully examined to confirm death.

I've added a statement: "Euthanasia was carried out through cervical dislocation, followed by confirmation of death through assessment of clinical signs, as per national legislation and internal policies", lines 122-124 in the Marked-Up document.

3. Line 155. Were no dilutions performed for the microbial count? Explain.

To clarify, we modified the statement as follows: "The resulting suspension was filtered using 70 µm cell strainers (Falcon), diluted (7X 10-fold dilutions on 96-well plates) and 2X 10µL drops have been plated onto selective and differential chromogenic medium (BD, ChromAgar MRSA II) to evaluate S. aureus bacterial load". This modification is now reported in lines 149-151 of the Marked-Up version.

4. Lines 176-185. This information can be included in the introduction or in Methods, leaving only the results for the Results section.

Thank you for the suggestion, but we prefer to leave the results as presented. In our opinion, this facilitates the comprehension of the results also to a reader that is not familiar with the specific topic.

5. Line 216. Add only the letter p. Correct the entire text and tables.

Thank you for the suggestion. We have replaced it to better indicate the p - value throughout the text and tables.

6. Lines 302-303. To which part of the immune response are these referring? Explain.

Thank you for this observation. Our statement was intended as a general hypothesis based on the observed reduction in bacterial load. We did not assess specific components of the immune response in our study and therefore cannot definitively attribute bacterial control to any particular immune mechanism. We acknowledge that the original wording may imply a more precise interpretation than our data support. To address this, we had revised the statement to make clear that it offers a general hypothesis of immune involvement rather than a conclusion about specific immune components. Old statement: "Then, the immune system was able to control the infection, and a steady-state condition was reached". New statement: "Then, it appears that the host defence mechanisms likely contributed to controlling the infection, leading to a steady-state condition", lines 303-305 of the Marked-Up document.

Re: Spectrum02642-25R1 (**Unravelling the mouse model of *Staphylococcus aureus* bacteraemia and sepsis: a systematic approach to better characterize host/pathogen interactions**)

Dear Dr. Emiliano chiarot:

Your manuscript has been accepted, and I am forwarding it to the ASM production staff for publication. Your paper will first be checked to make sure all elements meet the technical requirements. ASM staff will contact you if anything needs to be revised before copyediting and production can begin. Otherwise, you will be notified when your proofs are ready to be viewed.

Sincerely,
M. Victoria Delpino
Editor
Microbiology Spectrum